# MASKED VECTOR QUANTIZATION

## ABSTRACT

Generative models with discrete latent representations have recently demonstrated an impressive ability to learn complex high-dimensional data distributions. However, their performance relies on a long sequence of tokens per instance and a large number of codebook entries, resulting in long sampling times and considerable computation to fit the categorical posterior. To address these issues, we propose the Masked Vector Quantization (MVQ) framework which increases the representational capacity of each code vector by learning mask configurations via a stochastic winner-takes-all training regime called Multiple Hypotheses Dropout (MH-Dropout). On ImageNet 64×64, MVQ reduces FID in existing vector quantization architectures by up to $68\%$ at 2 tokens per instance and $57\%$ at 5 tokens. These improvements widen as codebook entries is reduced and allows for 7–45× speed-up in token sampling during inference. As an additional benefit, we find that smaller latent spaces lead to MVQ identifying transferable visual representations where multiple can be smoothly combined.

## 1 INTRODUCTION

In deep generative modelling, the choice of latent representation is an important consideration due to trade-offs in sample stability, quality, model size and compatibility with different modalities. Generative models with continuous latent random variables that assume parametric distributions have led the field in sample quality for the past decade (Vahdat & Kautz, 2020; Donahue & Simonyan, 2019). Despite their impressive performance, these models can be challenging to stabilise, resulting in problems such as posterior collapse (Lucas et al., 2019a; He et al., 2019; Lucas et al., 2019b).

Interest in discrete representations to address these challenges has seen a revival recently with the development of several discrete autoencoders (Van Den Oord et al., 2017; Razavi et al., 2019; Ramesh et al., 2021; Esser et al., 2021; Nichol et al., 2021; Rombach et al., 2022) with improved stability in high-dimensional visual and audio domains. This approach maps each instance to a discrete sequence of codebook indices (tokens) using vector quantisation (VQ) (Van Den Oord et al., 2017), Gumbel-softmax (Jang et al., 2016) or Concrete distributions (Maddison et al., 2016). In a secondary training stage, an autoregressive probabilistic model, such as a PixelCNN (Van Oord et al., 2016) or Transformer (Vaswani et al., 2017), learns the categorical posterior, representing the distribution of observable token sequences.

Despite these improvements, growing the representational capacity of discrete autoencoders remains tied to increasing the number of tokens assigned to each instance and the number of total codebook entries. Increasing both hyper-parameters can scale performance to high quality images as seen in Razavi et al. (2019) where the authors used 1280 tokens per image. However this resulted in unrealistically long ancestral sampling times and significantly higher computational resource use to fit the categorical posterior. More recent works, such as VQGAN (Esser et al., 2021) and Stable Diffusion (Rombach et al., 2022), introduced a discriminator (Goodfellow et al., 2014) and diffusion models (Sohl-Dickstein et al., 2015) to reduce token length to 256 per image however total sampling times remain rather long at 7258 seconds per image with a single Nvidia Titan X. Whilst demonstrating impressive quality, these times may continue to hinder the use of discrete autoencoders in challenging domains such as video generation and real-time applications particularly where compute resources are constrained.

To address this problem, we explore whether each instance can be compressed into a shorter sequence of tokens using a smaller codebook. We achieve this goal by introducing *Masked Vector Quantization*

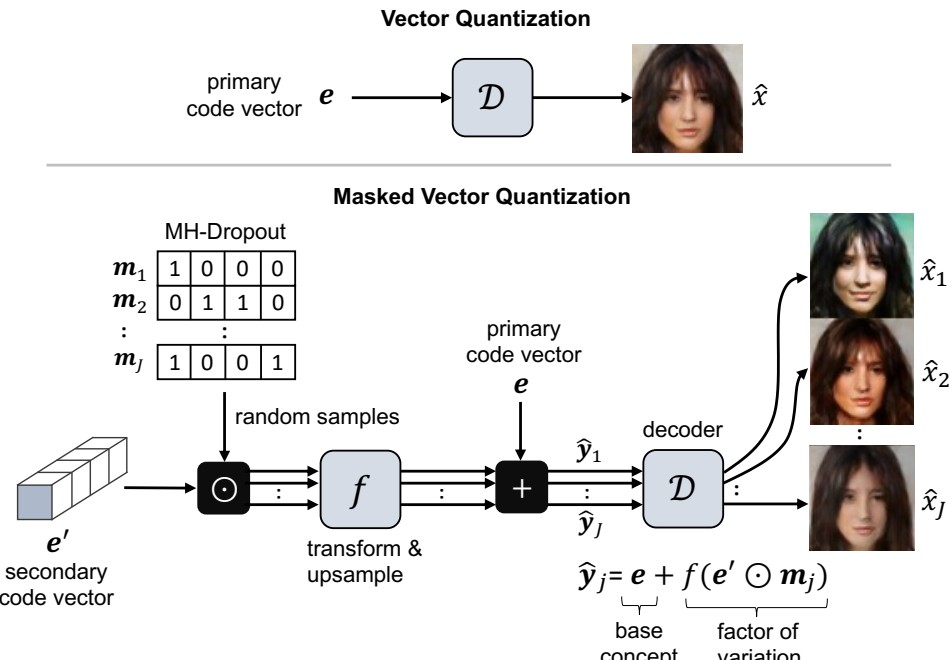

Figure 1: MVQ (bottom) compared to VQ (top). A unique pair of primary and secondary code vectors can encode up to $2^{D'}$ reconstructions. Multiple Hypotheses Dropout trains sampled mask configurations of the secondary code vector to represent different factors of variation (age, hair, skin tone, background, etc.). For example, by comparing $\mathcal{D}(\mathbf{e})$ to $\hat{x}_2$, the masked secondary code vector represents older age and make-up skin tone.

(MVQ), a novel variant of VQ that allows the masked configuration of each codebook vector to be individually mapped to separate instances. More precisely, in our framework each instance is encoded with three components: a primary code $\mathbf{e}$, a secondary code $\mathbf{e}'$ and a mask $\mathbf{m}_j$, as shown in Figure 1. In comparison to standard VQ, each primary code can represent a further $2^{D'}$ instances, where $D'$ is the embedding dimension of the secondary code.

During training, the best primary and secondary code vector for each instance can be found using standard nearest neighbor lookup from the codebook. It is, however, computationally infeasible to search for the best mask across all $2^{D'}$ possibilities in latent spaces of dimension typically used in practice. To overcome this, we introduce *Multiple Hypotheses Dropout* (MH-Dropout) a novel variant of dropout (Hinton et al., 2012) that incorporates multiple hypotheses training (Guzman-Rivera et al., 2012; Rupprecht et al., 2017). During the forward pass, $J < 2^{D'}$ masks are randomly sampled, yielding $J$ latent representations. During the backward pass, we use a *winner-takes-all* reconstruction loss where only the *best* of the $J$ representations affects the gradient.

The MVQ framework improves existing VQ architectures, such as VQ-VAE2 (Razavi et al., 2019) and VQGAN (Esser et al., 2021), particularly when the number of tokens per instance and codebook size is reduced. Across multiple medium resolution datasets, we observe FID reductions up to 82% at 2 tokens per instance, 57% at 5 tokens and 14% at 17 tokens. These improvements consistently grow as the number of codebook entries decrease and also reduce token sampling times by 58–87% on a consumer grade GPU and 80–97% on CPU.

We also find that the dimensions of masked codes learn characteristics that can be smoothly interpolated, combined and transferred to other primary codes. These features are sometimes clearly interpretable (such as the presence of eyeglasses or a smile) and can correspond to individual dimensions as seen in Figure 2.

Our paper is outlined as follows: In Section 2, we review related work on discrete autoencoders, dropout and multiple hypotheses training. In Section 3, we present background on the popular vector quantization framework which our work builds upon. In Section 4, we introduce our *Masked Vector*

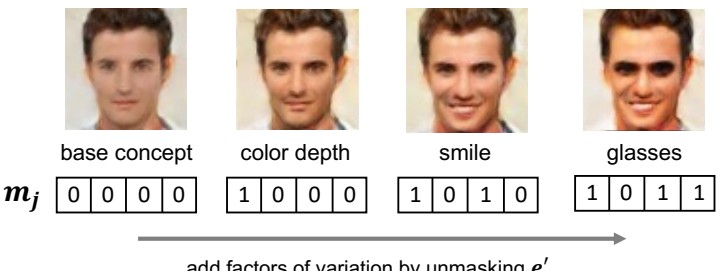

Figure 2: Each dimension of secondary code vector $\mathbf{e}'$ learns interpretable characteristics completely unsupervised. These characteristics can be controlled with mask $\mathbf{m}$ as seen above.

*Quantization* framework and *Multiple Hypotheses Dropout*. In Section 5, we describe the results of extensive experiments on our MVQ framework, benchmarking performance against both VQVAE-2 and VQGAN at multiple compression rates and datasets.

## 2 RELATED WORK

Our work is closely related to three areas of research: discrete autoencoders, dropout and multiple choice learning.

**Discrete Latent Models.** Autoencoders with discrete latent representations appeared as early as the work of Hinton & Zemel (1993), where the authors proposed learning a codebook with stochastic vector-quantization (VQ) in accordance with the Minimum Description Length (MDL) principle (Rissanen, 1987). Recent works have proposed the use of Gumbel-softmax (Jang et al., 2016) or Concrete distribution (Maddison et al., 2016) to learn codebooks with continuous reparameterization. In the work of Van Den Oord et al. (2017), the authors propose training codebooks with winner-takes-all VQ and an autoregressive model to learn the categorical posterior. Others have since combined codebooks with hierarchical latents Razavi et al. (2019), transformers (Ramesh et al., 2021), generative adversarial networks (GANs) (Esser et al., 2021) and diffusion models (Nichol et al., 2021; Rombach et al., 2021; 2022) to produce remarkable high quality data reconstructions.

However, contemporary discrete autoencoders come with a trade-off between compression and computational resources to learn the categorical posterior. Larger latent representations reduce reconstruction error, but require more computational resources and are slower to sample from, as outlined in Section 1. Our MVQ framework addresses this by learning a more efficient latent representations that uses MH-Dropout, a variant of binary dropout (Hinton et al., 2012).

**Dropout.** Binary dropout regularizes deep neural networks by randomly dropping hidden units to prevent co-adaption of features. This can be seen as a computationally tractable approach to bagging during inference by forming ensembles of sub-networks (Srivastava et al., 2014). In contrast to binary dropout, MH-Dropout actively recovers these representations during inference, leading to accurate multiple predictions by the ensemble. This increases compression as the same distribution is represented with significantly fewer parameters.

A similar approach using variational dropout (Kingma et al., 2015) appears in Molchanov et al. (2017), where the parameters of large supervised networks are significantly reduced, at a minor cost in accuracy. MH-Dropout samples masks during inference similar to Monte-Carlo dropout (Gal, 2016) and Concrete dropout (Gal et al., 2017). These, however, were primarily developed to quantify uncertainty for tasks in reinforcement and supervised learning.

**Multiple Choice Learning.** Multiple Hypotheses Dropout is inspired by work from multiple choice learning (MCL) which was first introduced by Guzman-Rivera et al. (2012) to address problems with multiple outputs given the same input. These approaches propose initializing multiple support vector machines (Guzman-Rivera et al., 2012) or multi-layer perceptrons (MLPs) (Rupprecht et al., 2017; Lee et al., 2017; Nguyen et al., 2021) to increase the number of predictors per input. MH-

Dropout takes a different approach, generating predictions by sampling from an ensemble of possible sub-networks which is significantly more scalable.

MCL techniques are commonly criticized for suffering from *modal collapse*. This problem arises when a subset of predictors are not trained due to poor weight initialization. MH-Dropout addresses this by back-propagating through a shared set of weights which also forces the network to adopt general features for multiple outputs.

## 3 MASKED VECTOR QUANTIZATION

Our solution extends the popular VQ autoencoder framework (Hinton & Zemel, 1993; Van Den Oord et al., 2017), so we begin this section by reviewing the concept. We then introduce our proposed architecture MVQ and MH-Dropout. For simplicity, throughout this section we use only one primary and one secondary token per instance when describing the algorithm. In practice, sequences of tokens are often used.

### 3.1 BACKGROUND: VECTOR QUANTIZATION

The winner-takes-all VQ framework utilizes a latent codebook containing a mixture of $K$ centroids $\mathbf{e}_k \in \mathbb{R}^D$ where $k = 1 \ldots K$. Given a dataset $X = (x_1, \ldots, x_n)$ of $n$ instances, this framework aims to map each instance to a discrete latent variable $z$ which refers to the codebook indices (tokens).

To achieve this, a primary encoder $\mathcal{E}$ is used to encode each instance to a vector $\mathbf{y}$. This vector is replaced with the indices of the nearest vector from the latent codebook, forming our discrete token $z$. The reparameterization process involves a simple lookup against the codebook indices which yields an equivalent vector representation $\hat{\mathbf{y}}$. The representation $\hat{\mathbf{y}}$ is then input to the decoder $\mathcal{D}$ which outputs a reconstruction of the instance $\hat{x}$.

The goal is to minimize the reconstruction loss between each instance and reconstructed output:

$$\mathcal{L}_{rec}(\mathbf{x}, \hat{\mathbf{x}}) = \|\mathbf{x} - \hat{\mathbf{x}}\|_2^2 \tag{1}$$

This is made possible by back-propagating through the codebook using the straight-through gradient estimator (Bengio et al., 2013) which passes the gradients directly from the decoder to the encoder with the following codebook loss:

$$\mathcal{L}_{cb}(\mathbf{y}, \hat{\mathbf{y}}) = \|sg[\mathbf{y}] - \hat{\mathbf{y}}\|_2^2 + \beta\|\mathbf{y} - sg[\hat{\mathbf{y}}]\|_2^2 \tag{2}$$

where $sg$ refers to the stop-gradient operation. This is commonly referred to as "commitment loss" (Van Den Oord et al., 2017). An additional stage is required to learn the categorical posterior distribution over the tokens using an autoregressive probabilistic model with cross-entropy loss.

A common approach to reducing both reconstruction and codebook loss is to increase the number of codebook entries $K$ or number of tokens per instance $S$. This scales representational capacity by decreasing the expected distance between $\mathbf{y} \approx \hat{\mathbf{y}}$, at the cost of training larger autoregressive models and longer sampling times. The MVQ framework scales representational capacity, but incurs these costs at a much lower rate as codebook size and number of tokens grow, as we verify in Section 4.

### 3.2 INCREASING REPRESENTATIONAL CAPACITY WITH MASKED VECTORS

Our MVQ framework builds upon the representational capacity of each latent VQ centroid by introducing a secondary code vector $\mathbf{e}' \in \mathbb{R}^{D'}$ and allowing its masked configurations to be mapped to each instance.

A masked configuration of a secondary code vector is defined as:

$$\mathbf{c} = \mathbf{e}' \odot \mathbf{m} \tag{3}$$

where $\mathbf{m} \in \{0, 1\}^{D'}$ is a binary mask and $\odot$ is an element-wise or Hadamard product.

Let $\mathcal{M} = \{\mathbf{m}_1, \ldots, \mathbf{m}_{J_{\max}}\}$ be the set of all $J_{\max} = 2^{D'}$ possible masks. If each mask in $\mathcal{M}$ is multiplied element-wise with the same code vector $\mathbf{e}'$, this constructs the set of all $2^{D'}$ possible masked configurations:

$$\mathcal{C} = \{\mathbf{c}_1, \ldots, \mathbf{c}_{J_{\max}}\} = \{(\mathbf{e}' \odot \mathbf{m}_1), \ldots, (\mathbf{e}' \odot \mathbf{m}_{J_{\max}})\} \tag{4}$$

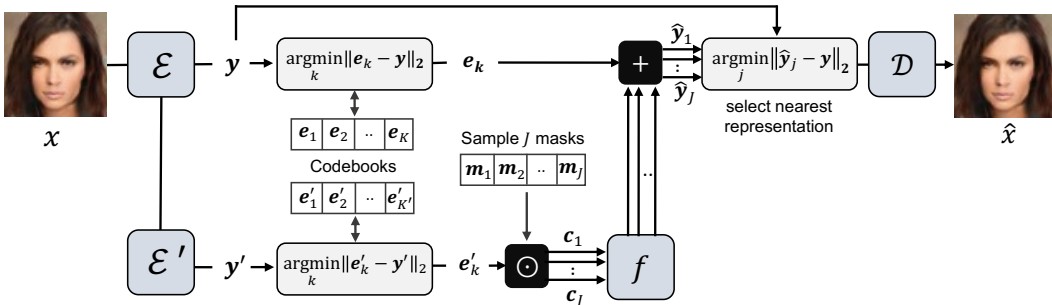

Figure 3: The MVQ architecture depicting the reconstruction of an instance during training. MVQ introduces a secondary branch to the VQ framework where a secondary code and $J$ random masks are sampled. Only the best of $J$ representations is passed to the decoder.

Thus in this framework, each instance is mapped from token to representation $\hat{\mathbf{y}}$ using two latent code vectors from separate codebooks and a binary mask from $\mathcal{M}$, $(\mathbf{e}, \mathbf{e}', \mathbf{m})$:

$$\hat{\mathbf{y}} = \mathbf{e} + f(\mathbf{e}' \odot \mathbf{m}) = \mathbf{e} + f(\mathbf{c}) \tag{5}$$

where $f$ is a non-linear multi-layer neural network. The complete architecture is illustrated in Figure 3 where the outputs of a secondary encoder $\mathcal{E}'$ is quantized into $\mathbf{e}'$ using a secondary codebook.

The training objective is to find the best combination of $(\mathbf{e}, \mathbf{e}', \mathbf{m})$ that minimizes the $L^2$-norm between $\hat{\mathbf{y}} \approx \mathbf{y}$. As the number of codebook entries is generally small, a simple codebook look-up is feasible for code vectors $\mathbf{e}$ and $\mathbf{e}'$. However when $D'$ is high dimensional, it is computationally impractical to enumerate through all possible $2^{D'}$ masks in $\mathcal{M}$.

To resolve this, we use MH-Dropout which approximates the best mask by using multiple hypotheses training. In this approach, only $J < 2^{D'}$ masks are randomly sampled from $\mathcal{M}$ and only the best mask which results in the nearest representation $\hat{\mathbf{y}} \approx \mathbf{y}$ is used during back-propagation. This algorithm is described in the following section.

### 3.3 MULTIPLE HYPOTHESES DROPOUT

MH-Dropout can be seen as efficiently converting $f$ into an ensemble of sub-networks $\{f_1, \ldots, f_J\}$ using the $J$ masked code vectors $\mathcal{C}$ as input. As seen in Figure 4, each sub-network results from the elimination of input connections by each mask, just as classic dropout sets connections to zero. Thus, sampling $J$ masks to find the nearest representation (in $L^2$ norm) to $\mathbf{y}$ is equivalent to sampling $J$ sub-network predictions (hypotheses) to find the nearest representation.

Only the *best* of the $J$ sub-network hypotheses is selected as the final representation and only the gradients of the respective *best* sub-network are back-propagated.

In the forward pass, $J$ masked code vectors are sampled from $\mathcal{C}$ and passed to $f$ to give $J$ sub-network hypotheses $\{f(\mathbf{c}_1), \ldots, f(\mathbf{c}_J)\}$. The best sub-network is the one with its hypothesis closest to vector $\mathbf{y}$ in $L^2$-norm.

Only this sub-network is addressed in the backward pass. Gradients for the other sub-networks are eliminated by multiplying their respective loss by zero, so the objective function is:

$$\mathcal{L}_{mhd}(\hat{\mathbf{y}}_{1:J}, \mathbf{y}) = \sum_{j=1}^{J} w_j L_2(\hat{\mathbf{y}}_j, \mathbf{y})$$
$$w_j = \delta(j = \arg\min_j \|\hat{\mathbf{y}}_j - \mathbf{y}\|_2) \tag{6}$$

where $w_j$ gates gradient of sub-networks by using $\delta$, which is the Kronecker-delta returning 1 for the best hypothesis and 0 otherwise.

**Multiple Hypotheses Dropout**

Figure 4: A forward pass through a MH-Dropout layer where $J$ random masks eliminate hidden connections, creating $J$ sub-networks that output $J$ hypotheses.

The overall MVQ objective function combines the MH-Dropout loss, reconstruction error and two VQ codebook losses:

$$\mathcal{L}_{MVQ} = \mathcal{L}_{rec}(\mathbf{x}, \hat{\mathbf{x}}) + \mathcal{L}_{cb}(\mathbf{y}, \mathbf{e}) + \mathcal{L}_{cb}(\mathbf{y}', \mathbf{e}') + \mathcal{L}_{mhd}(\hat{\mathbf{y}}_{1:J}, \mathbf{y})$$

(7)

This objective function can be flexibly modified to include a discriminator or perceptual loss (Johnson et al., 2016) as seen in Section 4.

## 3.4 DISCUSSION

MH-Dropout mitigates two related problems in multiple choice learning and discrete autoencoders. Multiple choice learning algorithms are commonly criticized for suffering from *modal collapse*. This occurs when a subset of predictors are not trained, due to poor weight initialization for example, and then used during inference. The same problem can be found in VQ autoencoders, when a subset of all possible token sequences are not trained. This is usually mitigated by introducing a probabilistic model to learn the categorical posterior over the used tokens, however it requires additional computational resources as token sequence length and number of codebook entries increase.

MH-Dropout avoids modal collapse because the shared set of weights are jointly trained in back-propagation. The sub-networks with hypotheses not closest to the primary representation will still be trained since their weights are shared with the winning sub-network. This forces the model to adopt a set of weights for $f$ and the secondary code vectors that generalize to multiple outputs. This is also why searching the ensemble of sub-networks by sampling masks is effective and creates a beneficial trade-off between computational resource use and finding the best mask (see Section 4.2).

We observe that degenerate outputs do not occur for any sampled mask applied to secondary code vectors during inference. It is therefore not necessary to learn a distribution over masks for generation (as is done for VQ-VAE codebooks), significantly reducing the size of categorical posterior models.

MH-Dropout has the binary dropout (Srivastava et al., 2014) property of tending to learn sparse secondary code vectors, with smaller numbers of larger values. Specific dimensions of the code vectors often correspond to distinct, interpretable characteristics such as a smile and glasses, as seen in Figure 2. These characteristics can be smoothly combined and transferred to other primary code vectors as we will demonstrate in the next section.

A similar approach to ours is product quantization (PQ) Jegou et al. (2010), which also creates multiple representation for each code vector. The difference is PQ breaks vectors into fixed partitions, whereas ours conducts a stochastic search of the projections of the vector using an mask operation.

## 4 EXPERIMENTS

Here we focus on understanding the impact of adopting MVQ and MH-Dropout by integrating them into two popular VQ architectures, VQVAE-2 and VQGAN. We discuss our experimental setup in Section 4.1. We investigate the effects of increasing sampled masks on MVQ quality in Section 4.2, and benchmark quality and reconstruction error across a wider range of datasets in Section 4.3. We then demonstrate that MVQ identifies features that can be transferred and combined in Section 4.4.

### 4.1 SETUP

**Model Comparisons.** Benchmarking generative models is a challenging task due to the wide range of factors that effect performance. To fairly assess MVQ we compare its performance to hierarchical VQ latents proposed in Razavi et al. (2019), and keep as many factors equivalent as possible. Thus, our comparative architecture to *VQVAE-2* simply replaces the top-bottom hierarchical VQ codebooks with MVQ, and is referred to as **MVQVAE**. Similarly, the hierarchical codebook version of *VQGAN* is compared to **MVQGAN**, but uses the same encoder and decoder from VQVAE-2 due to the GPU memory requirements of diffusion probabilistic models (Ho et al., 2020). The VQVAE2 models use a PixelCNN as the categorical posterior model, while the VQGAN models use a Transformer model as proposed in Rombach et al. (2021). Secondary codes are generated by conditioning on the sequence of primary codes. As previous works use different hyper-parameters and datasets, we re-train and evaluate their models based on open-source implementations.

**Hyper-parameters.** For each model, we vary two hyper-parameters to study the impact of masked code vectors: the number of codebook entries and tokens per image. There are a total of $K$ codebook entries, with each primary and secondary codebook having $K/2$ entries. The number of tokens per image is $S + S'$ where $S$ and $S'$ are the numbers of primary and secondary tokens respectively. Given an image $x \in \mathbb{R}^{h \times w \times 3}$, we extend previous work of Esser et al. (2021); Rombach et al. (2021) by studying a wider range of down-sampling factors, $\mathcal{F} = \sqrt{(h \times w)/S}$, from 7–64. Every set of comparative models are trained for equal epochs. For further hyper-parameters, see Appendix A.1.

**Datasets.** Experiments are conducted on medium resolution image datasets: FashionMNIST 28×28 (FMNIST) (Xiao et al., 2017), CIFAR10 32×32 (Krizhevsky et al., 2009), CelebA 64×64 (Liu et al., 2018) and ImageNet 64×64 Deng et al. (2009). Using the down-sampling factors provided above, the number of primary tokens per image are in the range 1–16. Due to this short length, we use 1 secondary token per image.

**Metrics.** Reconstruction and sample quality is assessed using Fréchet Inception Distance (FID) (Heusel et al., 2017) which measures similarity between real and generated samples. Reconstruction error is assessed using perceptual loss (LPIPS) for VQGAN based models. For all the above metrics, a lower value indicates better performance. F-score measures the coverage of diversity and quality in a single number as per the work of Sajjadi et al. (2018). Due to lack of space, sample FID (FID*), negative log-likelihood (NLL), F-score ($F_{\beta=1/8}$) are reported in Appendix A.2.

### 4.2 IMPACT OF INCREASING SAMPLED MASKS PER FORWARD PASS

To explore the effects of masked vectors on quality, we use the FashionMNIST dataset as it allows extensive testing of hyper-parameters due to its small size. Using VQVAE-2 as the baseline, multiple MVQVAE models are trained using increasing numbers of sampled masks per forward pass at high down-sampling factors of $\mathcal{F} = 28$ in Figure 5(a) and $\mathcal{F} = 14$ in Figure 5(b).

These results show that replacing hierarchical VQ with MVQ improves FID with as little as 16 sampled masks per pass. Increasing the masks per pass consistently improves reconstruction quality across all codebook and token lengths. As a result, MVQVAE with only 2 tokens per image outperforms several VQVAE2 models with 5 tokens which indicates better performance at higher

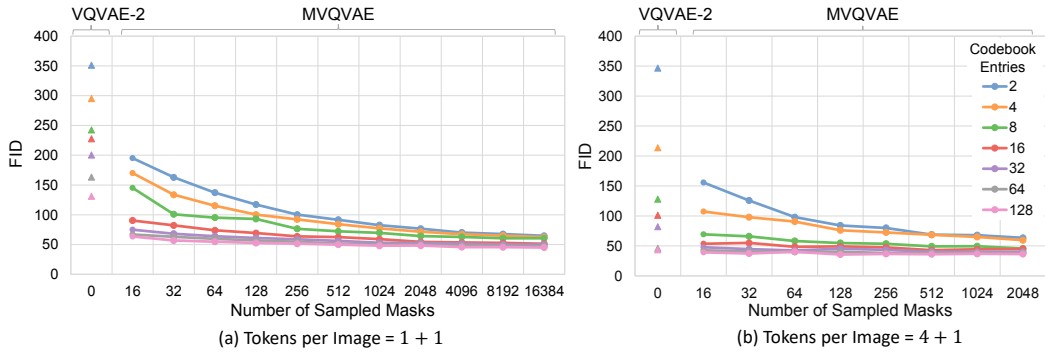

(a) Tokens per Image = 1 + 1

(b) Tokens per Image = 4 + 1

Figure 5: FID ↓ on FashionMNIST validation split. Increasing sampled masks per forward pass $J$ consistently improve FID for MVQVAE (ours). VQVAE-2 scores are shown left of each chart.

| Metric | Dataset | Tokens per Instance Codebook Entries | 1 + 1 | | 4 + 1 | | 16 + 1 | |
|---|---|---|---|---|---|---|---|---|
| | | | 16 | 256 | 16 | 256 | 16 | 256 |
| FID ↓ | CIFAR10 | VQGAN | 266.55 | 153.45 | 68.95 | 46.79 | 41.92 | 37.05 |
| | | MVQGAN | **58.78** | **57.22** | **46.70** | **41.19** | **36.73** | **34.84** |
| | CelebA64 | VQGAN | 113.54 | 87.67 | 36.07 | 28.78 | 26.47 | 24.28 |
| | | MVQGAN | **33.88** | **31.63** | **31.85** | **27.86** | **23.81** | **21.19** |
| | ImageNet64 | VQGAN | 285.94 | 184.30 | 161.19 | 77.95 | 56.67 | 47.99 |
| | | MVQGAN | **91.03** | **75.09** | **68.33** | **58.01** | **52.50** | **45.47** |
| LPIPS ↓ | CIFAR10 | VQGAN | 0.463 | 0.447 | 0.423 | 0.379 | 0.340 | 0.271 |
| | | MVQGAN | **0.406** | **0.390** | **0.377** | **0.342** | **0.322** | **0.266** |
| | CelebA64 | VQGAN | 0.345 | 0.327 | 0.288 | 0.251 | 0.230 | 0.186 |
| | | MVQGAN | **0.293** | **0.261** | **0.270** | **0.232** | **0.227** | **0.186** |
| | ImageNet64 | VQGAN | 0.558 | 0.537 | 0.531 | 0.488 | 0.466 | 0.406 |
| | | MVQGAN | **0.514** | **0.497** | **0.498** | **0.472** | **0.456** | **0.404** |
| $F_{\beta=8}$ ↑ | CIFAR10 | VQGAN | 0.132 | 0.744 | 0.770 | 0.882 | 0.874 | 0.906 |
| | | MVQGAN | **0.846** | **0.815** | **0.863** | **0.885** | **0.892** | **0.909** |
| | CelebA64 | VQGAN | 0.418 | 0.584 | 0.860 | 0.891 | 0.917 | 0.937 |
| | | MVQGAN | **0.868** | **0.881** | **0.861** | **0.915** | **0.932** | **0.952** |
| | ImageNet64 | VQGAN | 0.125 | 0.428 | 0.206 | 0.571 | 0.787 | 0.849 |
| | | MVQGAN | **0.557** | **0.620** | **0.703** | **0.752** | **0.832** | **0.872** |

Table 1: FID, perceptual loss and F-score ($F_{\beta=8}$) for MVQGAN ($J = 2048$) and VQGAN. MVQ outperforms VQ particularly when tokens per instance and number of codebook entries are reduced.

levels of compression. Despite sampling only a subset of all $2^{D'}$ possible masks, these results verify that MH-Dropout allows us to trade-off computational resource use as discussed in Section 3.4.

### 4.3 RECONSTRUCTION QUALITY

To understand scalability of performance, we conduct benchmarks across a wider range of datasets. VQGAN is used as the baseline as we find that the addition of perceptual loss and adversarial discriminator to VQVAE2 consistently improve metrics. Our proposed model, MVQGAN uses a fixed $J = 2048$ sampled masks per pass during training.

In Table 1, we present results which show that introduction of MVQ consistently improves FID, perceptual loss and F-score at all down-sampling factors $\mathcal{F}$ in the range 7–64. As seen in the previous section, improvement is greatest at reduced token lengths and number of codebook entries, which again shows that MVQ is more robust at higher compression rates.

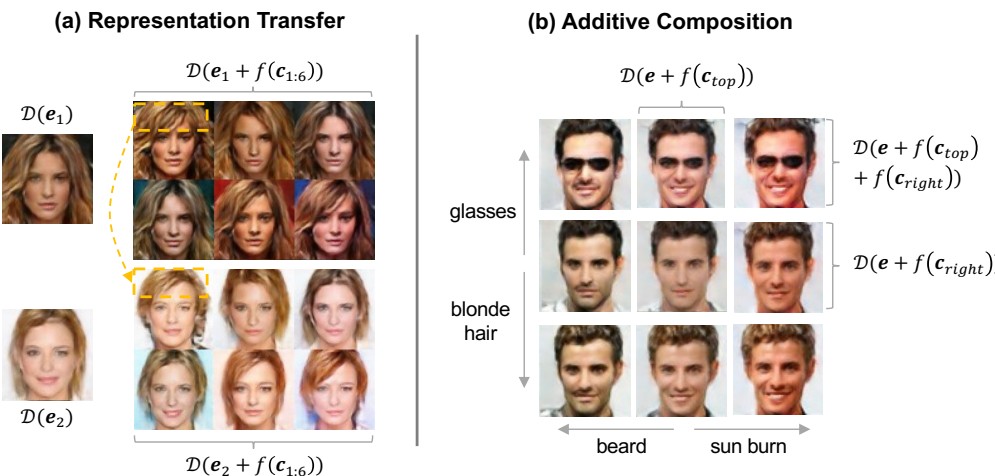

Figure 6: Samples generated by MVQGAN using only 16 codebook entries and $4 + 1$ tokens per instance. **(a)** Transfer of visual representations between two primary codes with different base styles. **(b)** Additive composition of multiple visual representations.

MVQGAN trained with only 2 tokens per image ($\mathcal{F} = 64$) on CelebA64 is reasonably competitive with VQGAN with 17 tokens ($\mathcal{F} = 16$). The same pattern is observed with all other datasets for MVQGAN with 5 tokens and 256 codebook entries in comparison to VQGAN with 17 tokens. These improvements allows us to trade a minor decrease in quality (see Appendix A.3 for images) for a more practical benefit of reducing sampling times during inference.

Increasing the down-sampling factor from 16 to 64 for 64×64 images would yield 7× speed-up in token sampling during inference on a consumer-grade Nvidia RTX2080, and 45× speed-up for an Intel i9 CPU as seen in the Appendix, Table 6. Similarly for 256×256 images, increasing the down-sampling factor from 16 to 32 would yield a theoretical 20–24× speed-up. These improvements would be useful for embedded models in edge devices where low inference times are required and in video generation or reinforcement learning research where sampling efficiency is low.

### 4.4 REPRESENTATION TRANSFER AND ADDITIVE COMPOSITION

We demonstrate that MVQ results in transferable visual representations that can be smoothly combined in output space. Figure 6a illustrates 6 masked configurations being transferred from primary code vector $\mathbf{e}_1$ to $\mathbf{e}_2$. Despite having different base styles, multiple visual representations (such as hair fringe, styles and colours) are smoothly transferred. Figure 6b depicts a grid of 9 images where the center image is the decoded primary code vector. Four sampled masked configurations are added to the centre and are shown in the orthogonal grid positions. The corner images are the result of summing the two masked configurations of their two nearest orthogonal images. These corner images demonstrate the smooth combination of multiple visual characteristics.

As these features are learned completely unsupervised, the choice of which features become transferable depends on the dataset. For example, on FashionMNIST we find that shades, patterns, width and arm lengths are transferable (see Appendix A.5). We also observe that MVQ models with reduced token lengths and codebook entries are better at transferability. This is because reducing primary VQ centroids increases the reliance on learning factors of variations with secondary masked codes.

## 5 CONCLUSION

In this paper we introduced the MVQ framework which allows masked configurations to be mapped to instances using a stochastic winner-takes-all training regime called MH-Dropout. MH-Dropout produces code vectors which avoid modal collapse and can learn sparse, transferable features. MVQ improves on both reconstruction error and sampling quality in comparison to VQ, and is considerably more robust when tokens per instance and codebook entries are reduced.

## 6 REPRODUCIBILITY STATEMENT

An anonymized codebase for ICLR reviewers has been uploaded to GitFront. The link is provided here. This codebase will be open-sourced after publication. To help others replicate our work, additional information on experiment hyper-parameters and metrics are found in Appendix A.1 and Appendix A.2 respectively.

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
