# OpenReview forum: "Masked Vector Quantization"
_ICLR.cc/2023/Conference — Submitted to ICLR 2023_

### Official Review · Reviewer_qWji · 2022-10-25

**Confidence:** 4
**Correctness:** 4
**Technical Novelty And Significance:** 2
**Empirical Novelty And Significance:** 2
**Recommendation:** 3

**Clarity, Quality, Novelty And Reproducibility:**

### Clarity, Quality, and Novelty

* If I understand correctly, the proposed method is partly based on the previous work VQ-VAE2[a]. The difference to VQ-VAE2 is the mask configuration added into the top-level codes. Although it shows improvement in the generation tasks with shorter sequences, the technical novelty seems limited.

### Reproducibility
Not enough implementation details are provided in the paper, for example, in VQ-VAE2, it trains with lots of tricks to address the issue of the codebook collapse: EMA to update the codebook, codebook reset, etc. I am not sure whether these are used in the paper. Personally, it would be hard to reproduce the experiments merely with the description available in the paper.

**Details Of Ethics Concerns:**

The paper focuses on a fundamental research problem, I don't have any ethics concerns on the paper.

**Strength And Weaknesses:**

### Strength
* [S1]: The approach is well presented, figures are clear and contain enough detailed information to present the approach
* [S2]: Generating long sequences is indeed computationally expensive (see supplementary material Table 6), the problem is well motivated
* [S3]: The related work is completed, and to the best of my knowledge, all the related fields are discussed.
* [S4]: Representations learned by MVQ with short sequences are interesting (Sec. 4.4)

### Weakness
* [W1]: Although the proposed method (MH-Dropout) shows improvement in image generation with shorter sequences, the reason why it works remains unexplained. I think the paper lacks a discussion on the intuition and the explainability of the proposed approach.

* [W2]: The proposed pipeline is conceptually similar to VQ-VAE2. In fact, in 4.1 Setup, the paper precises that it replaces the top-bottom hierarchical VQ codebooks in VQ-VAE2 with MVQ, which consists of injecting masks into the top-level codes if I understand correctly. In such a case, it would be better for the paper to discuss clearly the difference to VQ-VAE2 and clarify its own technical contribution.

* [W3]: The generation results are weak (Table 1, Figure 7). I am fully convinced that the community needs faster methods, however, I am not sure the community benefits from approaches that sacrifice too much on the generation quality.

* [W4]: The transformer part is not self-contained. It is not clear whether the primary and the secondary codes are in different scales and how to conduct generation for hierarchical codes.

[a] Razavi, Ali, Aaron Van den Oord, and Oriol Vinyals. "Generating diverse high-fidelity images with vq-vae-2." Advances in neural information processing systems 32 (2019).

**Summary Of The Paper:**

This paper presents Masked Vector Quantization (MVQ), which is an extension of hierarchical Vector Quantized Variational AutoEncoder (VQ-VAE [a]). MVQ introduces mask configuration on the secondary code vector and demonstrates its effectiveness on image generation with shorter sequences (for example, less than 20) per sample.

The paper uses a training schema named Multiple Hypotheses Dropout (MH-Dropout) to learn the mask configuration. The key idea is to sample a number of binary masks (with Dropout) and take the best one that minimizes the latent reconstruction.


[a] Razavi, Ali, Aaron Van den Oord, and Oriol Vinyals. "Generating diverse high-fidelity images with vq-vae-2." Advances in neural information processing systems 32 (2019).

**Summary Of The Review:**

As detailed in Section **Strength and Weakness**, I think improving the generation efficiency is important for autoregressive generation. The problem in the paper is well-motivated.

However, the current manuscript could be further improved. Particularly, I think it lacks a part to explain the intuition behind the proposed approach and why the proposed approach works. In addition, the difference to VQ-VAE2 is not clear and the experimental results are weak.

Considering the above concerns, my current rating is to reject the paper.

---

> ### Author Response · Authors · 2022-11-11
> **Authors' Response (1/2)**
>
> Dear Reviewer. Thank you for your useful feedback and comments. Please find below our responses.
>
> **Comment 1 [Reproducibility]:** Personally, it would be hard to reproduce the experiments merely with the description available in the paper.
>
> **Reply 1:** The code was anonymously open sourced at https://gitfront.io/r/user-5701462/4Gyb1cShKbGG/mvq/ as per the Reproducibility Statement on Page 10.
>
> ___
>
> **Comment 2 [W1]:** Although the proposed method (MH-Dropout) shows improvement in image generation with shorter sequences, the reason why it works remains unexplained.
>
> **Reply 2:** We have discussed the reason MH-Dropout is so effective in Section 3.4, but we can expand on it here.
> Two characteristics of the dropout mechanism (as described in Hinton et al (2012) [1]) are exploited by MH-Dropout.
> The first is that dropout exploits an ensemble of possible sub-networks.
> In the original binary dropout work [1], the approximate average of the ensemble predictions is used as the output during inference.
> MH-Dropout uses an approximate search mechanism to select the best performing sub-network at each training iteration to provide the optimal training signal for the other weights in the network.
>
> The second and related characteristic is that dropout encourages the network to learn a sparse set of weights (see Srivastava et. al (2012) [2] Section 7.1 and 7.2 for an explanation).
> The sparse weights are a better representation of the salient features in the training data, and allows these features to be directly accessed, as seen in Figure 2 of our paper.
>
> **References:**
>
> [1] Geoffrey E Hinton, Nitish Srivastava, Alex Krizhevsky, Ilya Sutskever, and Ruslan R Salakhutdinov. Improving neural networks by preventing co-adaptation of feature detectors. arXiv preprint arXiv:1207.0580, 2012.
>
> [2] Nitish Srivastava, Geoffrey Hinton, Alex Krizhevsky, Ilya Sutskever, and Ruslan Salakhutdinov. Dropout: a simple way to prevent neural networks from overfitting. The journal of machine learning research, 15(1):1929–1958, 2014.
>
> ___
>
> **Comment 3 [W2]:** Similarity/Differences with VQVAE2
>
> **Reply 3:** MH-Dropout is a general mechanism that improves representational capacity. We have chosen to focus on modifying VQ-VAE2 in the paper because VQ-VAE2 is the state of the art. MH-Dropout can also be used in supervised learning problems with more than one output per input (Multiple choice learning). This is a direction of future research.
>
> The novel contributions are outlined in Sec 3.2 and 3.3. of the paper, where in 3.2 we describe how each training instance is mapped to a primary and secondary code as well as a mask, and in 3.3 how multiple hypothesis dropout is used in training.
>
> Unfortunately, there is not enough space in the main paper to discuss both VQVAE1 and VQVAE2, but we have added  a paragraph to the Appendix highlighting the differences between MVQ and VQVAE2 (see Sec A.6).
>
> ___
>
> **Comment 4 [W3]:** The generation results are weak (Figure 7, Table 1).... approaches that sacrifice too much on the generation quality.
>
> **Reply 4:** We have focused the experimental results on VQ-VAE2, but with a lower sequence length and image resolution. Unfortunately, performing the experiments with the parameters used in [1,2] (1280 tokens per image and 256x256 resolution) was computational intractable with available resources. Instead, to isolate and study the impact of our contributions (MVQ and MH-Dropout), we performed comparative experiments with 2 - 16 tokens per image and a resolution of 64x64 pixels. Looking at the trend in  that the performance, we are confident the trend would continue to the parameters in [1,2].
>
> If fact we believe that much improved generation results would be obtained using diffusion probabilistic encoders and decoders.
>
> We should reinforce here that MVQ and MH-Dropout _improve_ generation quality relative to VQ-VAE2 at each set of token sequence length and image resolution we have experimented with. The approach does not sacrifice generation quality for efficiency, but improves it. As per Table 1, our proposal improves FID by 5-68\% on Imagenet, 3-70\% on CelebA and 6-78\% on CIFAR10 given the same number of tokens and codebook size.
>
> **References:**
>
> [1] VQVAE2 - Generating Diverse High-Fidelity Images with VQ-VAE-2, NeurIPS 2019
>
> [2]  Taming Transformers for High-Resolution Image Synthesis, CVPR 2021
>
> ___
>
> **Comment 5 [W4]:** Transformers detail is not self-contained.
>
> **Reply 5:** Details of the transformer implementation are found in Table 4 (Appendix) which indicates they are of the same scale. Generating hierarchical codes for transformers follows the same concept as VQVAE-2. The secondary codes are generated by conditioning on the sequence of primary codes. We have added this to the paper for clarity.

---

> > ### Author Response · Authors · 2022-11-11
> > **Authors' Response (2/2)**
> >
> >
> > **Comment 6 [Clarity, Quality, and Novelty]:** the technical novelty seems limited.
> >
> > **Reply 6:** We respectfully disagree because the technical novelty of MVQ is the proposed use of a semi-continuous representation  consisting of a primary and a secondary token that results in a higher representational capacity.
> >
> > To overcome the computational challenge of searching over all possible mask configurations, we proposed Multiple Hypotheses Dropout.
> > This novel technique approximates the posterior distribution over the masks which alleviates the need for another expensive categorical model.
> >
> > To our knowledge, there is no previous work that proposes these two contributions.

---

### Official Review · Reviewer_GRXC · 2022-10-25

**Confidence:** 4
**Clarity, Quality, Novelty And Reproducibility:** Some parts of this work is not clear …
**Correctness:** 3
**Technical Novelty And Significance:** 3
**Empirical Novelty And Significance:** 2
**Recommendation:** 3

**Strength And Weaknesses:**

#### **Strength**

 - The topic of designing a better vector quantization system is important, which has recently been adopted in a variety of tasks, including generative modeling.

 - Vector quantization regimes face a compression-fidelity trade-off, whereas a longer sequence reduces quantization error and increases modeling difficulty. This work proposes to overcome this issue through a novel VQ system.


#### **Weaknesses**

 - Despite including novel masking operations on the code, the motivation of this work is closely related to Product Quantization[1]. Discussion and comparison against relevant quantization methods should be provided.

 - Some details are not self-contained, including the designs of MVQGAN. For example, the authors refer to VQVAE-2 as the architectures of encoder and decoder, and "All architectures use a Transformer as the categorical posterior model as proposed in Rombach et al. (2021)", where **all** is quite confusing. The authors need to provide more details to make the work more self-contained.

The scope of experiments is unable to demonstrate the effectiveness of such modeling power.

 - Since the major target of vector quantization is not to do compression, the reconstruction quality, errors, or sampling speed, cannot reflect the learned token representations as reliable. In fact, there exist many adaptations that reduce quantization error, while resulting in a worse token representation that might be more difficult for the probabilistic model to learn.

 - Applying an aggressive downsampling rate such as $f$=16 for small images used in the experiments makes less sense, as the obtained sequence length is already short, e.g. 4x4 . A shorter sequence and extremely small codebook hinders the contribution of this work, of which a non-auto-regressive is able to model well(e.g. MaskGIT[2] for 16x16 tokens on ImageNet).



[1] Product quantization for nearest neighbor search. T-PAMI 2011
[2] MaskGIT: Masked Generative Image Transformer. CVPR 2022

**Summary Of The Paper:**

 - This work proposes a Masked Vector Quantization (MVQ) framework to boost per-code capacity, targeting at representing images with shorter sequence or fewer codebook entries.

 - In the framework of MVQ, each instance is mapped into three components, including a primary code, a secondary code and a mask. The computational infeasibility is mitigated by Multiple Hypotheses Dropout.

 - The authors demonstrate that with MVQ, improvements on FIDs over VQGAN are observed, especially when the number of tokens per instance and codebook size is reduced. The authors also show some properties of learned codes.

**Summary Of The Review:**

 - Although this work is studying this important question of vector quantization, lack of solid experiments, such as the feasibility of training a better generative model with this MVQ system, and improvements on longer sequences and larger dataset, are unable to show the effectiveness of this method. Note that the target of VQ is not to solely do better compression, but for handling complex signals that existing models are unable to do well.

---

> ### Author Response · Authors · 2022-11-11
> **Authors' Response (1/2)**
>
> We appreciate the reviewer's useful feedback and and thank them from their comments. Please find below our responses.
>
> **Comment 1:** Despite including novel masking operations on the code, the motivation of this work is closely related to Product Quantization. Discussion and comparison against relevant quantization methods should be provided.
>
> **Reply 1:** Regarding Product Quantization: Thank you for this suggestion. We have amended the Section 3.4 to include this comparison. We provide a longer version here due to limited space.
>
> Product quantization (PQ) is primarily a multimedia retrieval technique and has not been applied to generative modelling yet as far as we know. PQ is similar to MVQ in the sense that it creates multiple representations for each training vector.
> The main differences between the two techniques are that PQ breaks a training vector into fixed size and location partitions, where MVQ conducts a stochastic search of the projections of the secondary latent vector through the masking operation.
> PQ would not likely have the impact on training of the encoder and decoder that MVQ and MH-Dropout has, where the winner-takes-all selection after masking provides an optimal training signal.
>
> ___
>
> **Comment 2:** Some details are not self-contained. All architectures use a Transformer as the categorical posterior model as proposed in Rombach et al. (2021)", where all is quite confusing.
>
> **Reply 2:** Thanks for this suggestion. Many experiments details are included in the Appendix A due to lack of space.  If it helps, the code is anonymously open sourced at https://gitfront.io/r/user-5701462/4Gyb1cShKbGG/mvq/ as per the Reproducibility Statement.
> We clarified the explanations in the experiment section to address this.
>
> - "The VQVAE2 models use a PixelCNN as the categorical posterior model, while the VQGAN models use a Transformer."
> - For fairness of comparison, all models outlined in experiment section (MVQVAE, VQVAE2, VQGAN, MVQGAN) used the same hyper-parameters and encoder/decoder.
> - The reason for this is to isolate the impact of adopting MVQ and MH-Dropout.
>
> ___
>
> **Comment 3:** Experiments unable to demonstrate effectiveness of modelling power.
>
> **Reply 3:**
> It is unclear what benchmarks are being requested to demonstrate modelling power. We have tried to conform to the comparison approach and metrics used in the recent papers that we think should be the most appropriate comparisons:
>
> * VQVAE1 [1] - FID, Reconstruction Error
> * VQVAE2 [2] - FID, Inception Score, Reconstruction Error, Precision, Recall
> * VQGAN  [3]  - FID, Inception Score, Reconstruction Error
>
> We have included all of these except inception score.
>
> Could the reviewer also please clarify what is meant by "cannot reflect the learned token representations as reliable"?
>
> ___
>
> **Comment 4:** Applying an aggressive downsampling rate such as f=16 for small images used in the experiments makes less sense, [...] A shorter sequence and extremely small codebook hinders the contribution of this work.
>
> **Reply 4:**
> The paper argues that masked code vectors result in representational capacity improvements and verifies this with experiments over multiple sequence lengths and codebooks. We do not think that shorter sequences are always better, nor is this the main contribution.
>
> We have tried, in the experiments, to show the effect of MVQ and MH-Dropout at a range of sequence length and image resolution parameters, including at the extremes. Unfortunately, we do not currently have access to the computational resources to extend these experiments to the large end of the range, but we believe the trend is evident in the results.
>
> Given the same number of tokens and codebook size, our contribution improve representational capacity with an efficient mechanism that provides greater quality.
> Note that in the VQ autoencoder framework, higher reconstruction quality comes at the cost of increased number of tokens and codebook size (reducing compression).
>
> We believe that investigations of this sort are as important as studying scale from a research perspective.
> If the community continues to encourage only larger models, it will increase the cost of entry, lock out large portions of the research community and lead to slower progress over the long run.
>
> We would also like to point out that, as discussed by the reviewer, longer sequences and larger codebooks require a larger categorical posterior model, with corresponding increases in training time and computational resources.
> This was empirically shown in Table 4 where parameter count was reduced from 9.5M to 0.2M (over 97\%) by reducing sequence and codebook size.
>
> **References:**
>
> [1] Discrete Neural Representations, NeurIPS 2017
>
> [2] VQVAE2 - Generating Diverse High-Fidelity Images with VQ-VAE-2, NeurIPS 2019
>
> [3]  Taming Transformers for High-Resolution Image Synthesis, CVPR 2021

---

> > ### Author Response · Authors · 2022-11-11
> > **Authors' Response (2/2)**
> >
> > **Comment 5 [Clarity, Quality, Novelty And Reproducibility]:** Some parts of this work is not clear and hard to follow.
> >
> > **Reply 5:** We are happy to address this if the reviewer can be more specific.
> >
> > **Comment 6:** In fact, there exist many adaptations that reduce quantization error, while resulting in a worse token representation that might be more difficult for the probabilistic model to learn.
> >
> > **Reply 6:** Could the reviewer kindly provide references to these adaptions and proof that "they are more difficult for the probabilistic model to learn"?
> > While this may possibly occur, this anecdotal evidence is insufficient to act upon during a rebuttal period.

---

### Official Review · Reviewer_6yNV · 2022-10-26

**Confidence:** 5
**Correctness:** 3
**Technical Novelty And Significance:** 4
**Empirical Novelty And Significance:** 4
**Recommendation:** 10

**Clarity, Quality, Novelty And Reproducibility:**

The manuscript is well written, the quality of method , experiment and images are high , the method is very novel and , more importantly , clear. The work is original and needed by the community urgently.



**Strength And Weaknesses:**

The authors solve a real problem and pain point confront by many in discrete token based generative model for vision task, using a semi-continuous represention consisting of a primary and a secondary token. They empirically show using different generative task to show their approach improves performance from different aspects. The problem setting is clear, the approach at a high level is great and the results are solid

However, the reason why the dropout mask and multi-hypothesis version is use is not probably reasoned. There are many alternative possibilities including a small Bayesian neural network , learnable dropout such as Contextual Dropout or GFlowOut. The reason to use the current masking method to be properly explained. I understand it does not have to be the most sophisticated method in the literature but at least the author need to explain why it cannot be simpler.

**Summary Of The Paper:**

As a researcher working in the similar field, I definitely appreciate the author's efforts to solve a long-standing problem faced by many using a method that is reasonable according to the nature of the discretization process. The author seek to solve the problem that long sequences of discrete making it very difficult for transformer or other method to model in an autoregressive manner, which leads to difficulty in generative task. The author use a non-linear combination of two set of tokens , the primary and the secondary , to make the output representation semi-continuous and with higher capacity, using a MLP projector and dropout mask. This strategy can be intuitive understood as estimating the posterior of representation distribution given the latent consisting of two different tokens ,in a mathematically non-tight way.

**Summary Of The Review:**

This is a great piece of work due to its timing in solving an urgent problem in the research field as well as in the industry. The quality of the work is high , the method is novel and clean. One thing I like to point out is that figure is difficult to understand , not sure it is helping or making it more difficult for the readers. I would suggest improve it

---

> ### Author Response · Authors · 2022-11-11
> **Authors' Response**
>
> Dear Reviewer, thank you so much for your great feedback. We really appreciated your positive comments and completely agree that this paper tackles a very important problem with the adoption of discrete autoencoders. We also agree that the proposed solution is relevant to a large proportion of the community, both research and industry.
> Below you will find replies to your comments.
>
> **Comment 1:** The reason why the dropout mask and multi-hypothesis version is use is not probably reasoned. There are alternative possibilities such as Bayesian networks, learnable dropout such as Contextual Dropout or GFlowOut.
>
> **Reply 1:** Thank you for interest and question.
> In a parallel experiment, we attempted learnable dropout however it did not work well.
> The reason was because the Bernoulli probability vector, representing the probability of masking an individual dimension, converged to an equilibrium where parts of $\mathcal{M}$ (the set of all possible masks) are not explored well (i.e. when p is too low or high).
> This causes the model to either ignore or over-use code weights which resulted in many degenerate samples during inference.
>
> Our proposed stochastic search forces the network to explore all possible mask configurations and weight combinations equally.
> This also forces the codebook vector to adopt weights which generalize across many instances and resolved the degenerate sample problems.
>
> Multiple hypotheses training was the main ingredient needed to achieve this goal because it enables the network to map multiple outputs to the same codebook vector.
>
> In addition, many of the existing dropout methods are tailored towards uncertainty estimation or regularization, whereas our motivation was to increase the capacity of code vectors.
> I hope this answers your question.
>
> ___
>
> **Comment 2:** Figure is difficult to understand.
>
> **Reply 2:** We are happy to make changes to improve the clarity of the figure. Are you able to clarify which figure you are referring to?

---

### Decision · Program_Chairs · 2023-01-20

**Decision:**

Reject

**Justification For Why Not Higher Score:**

The significance is not high due to limited experimental study in the regime of small number of tokens.

**Justification For Why Not Lower Score:**

N/A

**Metareview: Summary, Strengths And Weaknesses:**

This paper presents Masked Vector Quantization (MVQ), which is an form of of hierarchical Vector Quantized Variational AutoEncoder (VQ-VAE. The paper uses a training schema named Multiple Hypotheses Dropout (MH-Dropout) to learn the secondary codebook. The key idea is to sample a number of binary masks (with Dropout) and take the best one that minimizes the latent reconstruction.

Strength
The problem setup is well-motivated and presentations are clear.

Weakness
The paper lack a stronger theoretical justification on the multiple hypotheses learning approach. While this could be potentially made up by empirical evidence, we do not see strong results demonstrating the method in challenging scenarios. In fact, we think the number of tokens used in experiments are too small, resulting in poor generation quality.

**Summary Of Ac-Reviewer Meeting:**

We had a virtual meeting on this paper (with Reviewer 6yNV not present). Main notes are:

This is a paper about learning discrete representation for images. Audiences in this community are mainly concerned with generation quality. The paper currently does not present high generation quality, as experiments are done with a small number of tokens. The reviewers think the experiments should be extended to more tokens with reasonable computational resources (few GPUs). On the other hand, there are not enough empirical results for claiming benefits in generation efficiency or compression.

In terms of methodology, the idea of using a secondary discrete representation with binary codes could be interesting and somewhat novel, but the benefit is not clearly demonstrated in the current paper, due to the overall poor generation results.

So we will recommend rejection for this paper.